# Revisiting the invasion paradox: Resistance-richness relationship is driven by augmentation and displacement trends

**Yu Zhu**, **Babak Momeni***

Biology Department, Boston College, Chestnut Hill, Massachusetts, Unites States of America

* momeni@bc.edu

## Abstract

Host-associated resident microbiota can protect their host from pathogens—a community-level trait called colonization resistance. The effect of the diversity of the resident community in previous studies has shown contradictory results, with higher diversity either strengthening or weakening colonization resistance. To control the confounding factors that may lead to such contradictions, we use mathematical simulations with a focus on species interactions and their impact on colonization resistance. We use a mediator-explicit model that accounts for metabolite-mediated interactions to perform *in silico* invasion experiments. We show that the relationship between colonization resistance and species richness of the resident community is not monotonic because it depends on two underlying trends as the richness of the resident community increases: a decrease in instances of augmentation (invader species added, without driving out resident species) and an increase in instances of displacement (invader species added, driving out some of the resident species). These trends hold consistently under different parameters, regardless of the number of compounds that mediate interactions between species or the proportion of the facilitative versus inhibitory interactions among species. Our results show a positive correlation between resistance and diversity in low-richness communities and a negative correlation in high-richness communities, offering an explanation for the seemingly contradictory trend in the resistance-diversity relationship in previous reports.

## Author summary

Empirically, different trends have been observed between the richness of a resident community and how resistant that community is against the introduction of new members: resident communities with higher richness can be more resistant against invaders in some cases and less resistant in other cases. To explain what can cause such seemingly contradictory trends, we used simulations of the invasion process using a simple model of microbial interactions through metabolites and metabolic byproducts. We found that two underlying trends consistently exist: the chance of augmentation decreases and (invader species added, without driving out resident species) and the chance of displacement

**Data Availability Statement:** All codes related to the results presented in this manuscript are available at https://github.com/bmomeni/invasion_richness.

**Funding:** YZ was supported by an Undergraduate Fellowship from Boston College. BM was supported by NSF-CBET (Grant No. 2103545). The funders had no role in study design, data collection and analysis, decision to publish, or preparation of the manuscript.

**Competing interests:** The authors have declared that no competing interests exist.

(invader species added, driving out some of the resident species) increases, in communities with higher richness. The combination of these two trends can lead to a not monotonic relationship between resistance against invaders and the species richness of the resident community. Our model predicts that in low-richness (versus high-richness) communities, resistance-diversity has a positive (versus negative) correlation, explaining the previously observed trends in the resistance-diversity relationship.

## Introduction

Host-associated microbiota co-evolve with their host and benefit the host through sharing metabolites [1,2], promoting development [3,4], and importantly, resisting pathogens [5,6]. Microbiota's ability to resist the invasion of non-native species is termed "colonization resistance" [7–9]. The strength of colonization resistance is influenced by the niche environment (e.g., temperature, salinity, and resource availability) [10–12], properties of the resident community (e.g., diversity, population size, microbial interactions) [13–15], traits of the potential invader (e.g., growth rate and dispersal compatibility) [16], and the way an invader interacts with the resident community (e.g., competition, parasitism, or antagonism) [15,17,18]. The underlying mechanisms for microbial resistance to invader colonization remain unclear because many influential factors can be involved and the relative importance of such factors is not known [18].

Prior work has focused on the impact of community diversity on invasion, because diversity plays a crucial role to maintain the multifunctionality of ecosystems [19–21]. However, the relationship between the diversity of a microbial community and the strength of its colonization resistance remains inconclusive. This is referred to as *invasion paradox* in community ecology, stating that observational or experimental evidence supports both the negative and positive relationships between the biodiversity of native species and the invasion of exotic species [22].

From one perspective, a positive correlation between microbial diversity and colonization resistance originates from Elton's observation that simple ecosystems (e.g., oceanic island and crop monoculture) are more vulnerable to invasions [23,24]. Furthermore, many classic ecologists in the 1960s and 1970s supported the same belief based on niche theory [17,23,25]. Niche theory defines the availability of niche opportunities as the condition that promotes invasion. This opportunity could take different forms, from resource availability to the absence of natural enemies. In this context, with high species diversity fewer niche opportunities will be available [17,24,26–29]. The same idea is also framed as 'Ecological Controls' (EC) or niche filling in other literature, with a similar concept that at higher richness fewer niches will be available for invaders, leading to stronger resistance [30]. Such a trend is observed in some microbial communities as well [14,29,31,32]. For instance, the wheat rhizosphere community had a decreased invasibility by the opportunistic pathogen *Pseudomonas aeruginosa* with an increased level of diversity controlled by dilution extinction gradient [31]. Similarly, when the grassland soil microbial community is invaded by *Escherichia coli*, *E. coli*'s survival is negatively correlated to microbial diversity [14].

A negative correlation between microbial diversity and colonization resistance has also been observed in various ecosystems. In some natural ecosystems, including riparian plant communities along rivers [33] and plant species at the Hastings Reservation [34], a positive relationship between native and exotic species diversity is observed. In experiments of introducing invaders into the grassland community, more invasive species were observed on the

herb-sown plots with more resident species [35] and the species-poor plots were generally not invaded [36]. This idea is sometimes framed as diversity begets diversity (DBD) [37]. The reason for these observations remains unexplored. One possible explanation is that a diverse community creates more distinct microenvironments that can support invading species. However, this explanation would predict the invader composition of high richness cases to be similar to that of the average of low richness cases, but this prediction was not observed in some cases [38].

There have been attempts to provide explanations for the invasion paradox. Levine [39] concludes that the positive correlation between diversity and resistance appears at a small scale (e.g. controlled experiments) while the negative correlation between diversity and resistance is observed at the community level. Their explanation is that factors covarying with diversity such as disturbance, propagule pressure, and species composition are responsible for the observed trends [23,39]. There are also additional factors which may make it hard to interpret the observational studies. In some studies, the fraction of exotic species in the community is used as a measure of invasibility [33,34]; however, this may be different from the response of the community to invasions, because the richness of the native community may be affected during the process of invasions [23,35]. Furthermore, the invasion process includes an initial introduction step [13], which can be confounding in some experiments while the probability of introduction of an invader is uncontrolled [36].

Since many potential factors can lead to positive versus negative correlation between diversity and resistance, we use mathematical models to minimize confounding factors. Previous studies have investigated the correlation between the diversity and invasibility of the community using *in silico* communities constructed based on pairwise Lotka-Volterra (LV) models [28,40–43]. The pairwise interactions between species are defined either through interference competition [41] or through competition for resources [28,43]. These studies discover that more complex communities (i.e., communities with more species so more interactions) are more vulnerable to invasions. However, since pairwise LV models may not adequately capture microbial interactions and higher order effects [44], we use a mediator-explicit model [44,45] to track both species and interaction mediators, mediators for short, which are compounds (such as nutrients, metabolic by-products, and toxins) that mediate the interactions between species. Such a model accounts for direct and indirect interactions through mediators in the shared environment. Mediators are produced and consumed by species and can influence species growth positively (as facilitators) or negatively (as inhibitors). We note that the mediator-explicit model is similar to, but distinct from, the standard consumer-resource model (CRM) [46,47], in that in CRM only beneficial resources are included and negative interactions only arise from competition for such resources. Moreover, while the majority of previous studies only focus on the success of the invader, we examine the fate of both the invader and the resident community as a function of resident community diversity. The motivation is to resolve some of the ambiguity around the relationship between diversity and invasion of the microbiome community in previous reports.

Before presenting our results, we should clarify the scope of this work. First, the term diversity has been used in the literature for various purposes, including taxonomic diversity (i.e., species-level analysis), functional diversity (i.e., substrate utilization analysis), and genetic diversity (i.e., nucleic acid analysis) [48]. We focus on taxonomic diversity in the form of species richness, with each species defined by its distinct growth properties and pattern of interactions. Second, our model parameters do not strictly represent specific real-world microbial species. Thus, our simulations do not capture any particular microbial community. Instead, they provide general insight of possible trends in a controlled setting. Lastly, our microbial communities have a well-mixed environment [44], and our investigations strictly assess alpha diversity (i.e., intracommunity diversity) [49,50].

Our results show that as the richness of the resident community increases, often the chance of colonization resistance shows an increase in low richness communities and a decrease in high richness communities. This effect is largely caused by two underlying trends at higher richness: an increase in instances where the invader displaced one or more members of the resident community and a decrease in instances where the invader is added to the resident community without driving any of the resident species to extinction. We consistently observe these trends at a range of parameters explored in this investigation.

## Result

### Resistance is often strongest at intermediate richness of resident communities

We simulate the *in silico* microbial communities using the mediator-explicit model (Eq 1). Mediators are the chemical compounds present in the environment that can interact with species in four possible ways: species either produce or consume mediators and mediators either facilitate or inhibit the growth of species [15,44,45]. Specifically, the rate of change of species $i$'s concentration is proportional to the cell densities of the species ($S_i$) scaled by its basal growth rate ($r_{i0}$) in addition to the sum of mediator $j$'s influence, which is the product of interaction strength on species $i$ ($r_{ij}$) and the concentration of mediators ($M_j$) controlled by the saturation concentration ($K_{ij}$) (Eq 1B). Similarly, the rate of change of mediators $i$ is total results of individual species $j$ if they produce the mediator $i$ with a rate of $\beta_{ij}$ or consume it with a rate of $\alpha_{ij}$ (Eq 1A).

$$\frac{dM_i}{dt} = \sum_j \left[ \beta_{ij} - \alpha_{ij} \frac{M_i}{M_i + K_{ij}} \right] S_j \tag{1A}$$

$$\frac{dS_i}{dt} = \left[ r_{i0} + \sum_j r_{ij} \frac{M_j}{M_j + K_{ij}} \right] S_i \tag{1B}$$

Using the mediator-explicit model, we first generated stable resident communities with multiple species *in silico*. For this, we assembled interacting species with random interactions and simulated the dynamics until a subset of species reached stable coexistence. In these simulations, we investigated initial pools of species with different ratio of positive versus negative mediator influences (+:- = 10:90, 50:50, or 90:10) to examine how the invasion trends may change if the interactions among species is shifted from having more facilitation (more "positive" influence of mediators on species) to more inhibition (i.e. more "negative" influence of mediators on species). We then introduced the invader into these resident communities and simulated the dynamics to assess the invasion outcomes (Fig 1, top). We categorized the stabilized community to four outcomes based on the presence of the invader and the change of richness of the resident species [15]: resistance (all resident species survive, invader goes extinct), augmentation (all resident species survive, invader survives), displacement (some resident species extinct, invader survives), and disruption (some resident species extinct, invader goes extinct) (Fig 1, top).

We observed that as the species richness of the resident community increased, resistance as the invasion outcome increased for low-richness resident communities and then decreased for high-richness ones (Fig 1, bottom). This was driven by an increase in instances of displacement, despite a decrease in instances of augmentation. In these cases, the chance of disruption remained low as species richness increased. Even though the trend of resistance is often of interest from an ecological perspective, when the richness of resident communities increases,

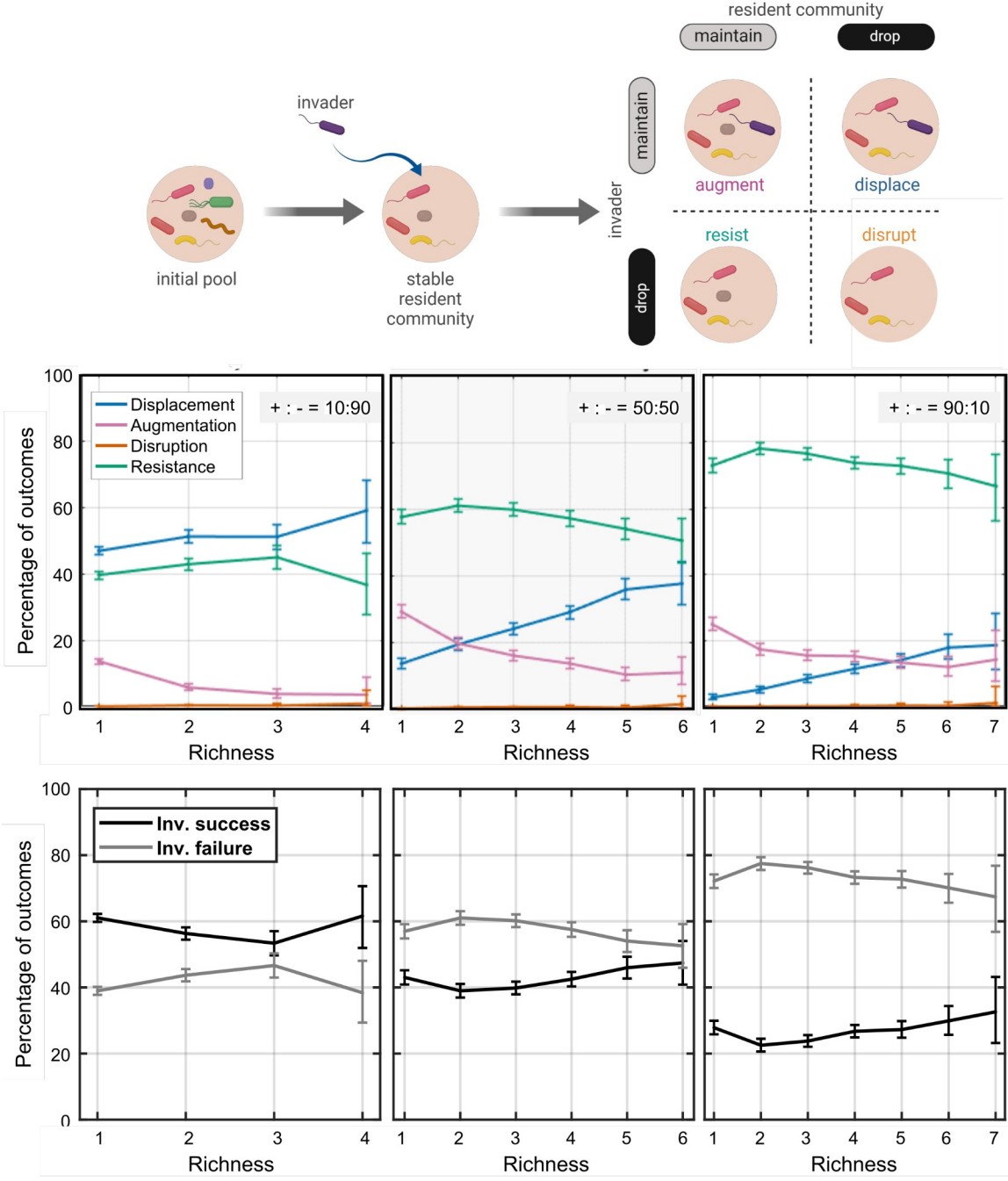

**Fig 1. Increased richness of resident communities leads to higher chance of displacement and lower chance of augmentation.** Top panel shows the overall invasion assay used throughout this manuscript: an initial pool of randomly interacting species is simulated until it reaches stability (200 generations), then an invader is introduced and the invasion outcome is assessed after 200 generations of growth through rounds of growth and dilution. In the middle panel, the relationship between richness and different outcomes is examined when the initial pool of resident species has more inhibitory interactions (+:- = 10:90), equal mix of inhibitory and facilitative interactions (+:- = 50:50), and more facilitative interactions (+:- = 90:10). In the bottom panel, the outcomes are grouped based on the success or failure of the invader. For each plot, 10,000 instances are examined. Interactions between resident members and the invaders are equal mix of inhibitory and facilitative interactions. The error bars show 95% confidence level estimated assuming a binomial distribution for each outcome. Only data points (at each richness value) are shown that had a sample size greater than 30. Generated using BioRender.com.

the consistent underlying trends are a decrease in the augmentation chance and an increase in the displacement chance. This may explain some of the seemingly contradictory trends reported in the past.

The measurement of species diversity in our investigations has been based on species richness. We examined if the observed trends applied to other measures of diversity [50]. We measured the species evenness using the Shannon index [50]. We found that the Shannon index of the community showed a complex trend: at any given richness, resident communities with higher evenness showed lower assimilation, disruption, and displacement; however, as the richness increased, augmentation increased (Fig A in S1 Text). To simplify interpretations, in the rest of this manuscript we use richness as the main measure of diversity.

To assess whether similar trends are observed under different interaction models, we repeated the invasion simulations in Fig 1 using a Lotka-Volterra (LV) model. We observed that in the LV model, similar to the mediator-explicit model, as the richness of the resident community increased the chance of displacement increased and the chance of augmentation decreased, leading to a nonmonotonic relationship between resistance and richness (Fig B in S1 Text). This pattern was observed consistently over a range of values for the mean and the spread of interaction coefficients (Figs B and C in S1 Text).

## More facilitative interactions lead to more augmentation and fewer displacements

We analyzed the relationship between community resistance with species richness, under the conditions that the microbial interactions between the resident members were more inhibitory, equally facilitative or inhibitory, or more facilitative (Fig 1, bottom). To investigate the effect of interactions among resident species, we examined the invasion outcomes at three different frequencies of facilitative interactions among species in the initial pool of species (i.e. three different +:- ratios; Fig 1, bottom). Generally, the displacement outcome was more likely when the microbial interactions were less facilitatory and more inhibitory. In contrast, the disruption outcome—even though generally rare under these conditions—was more likely when the microbial community had more facilitative interactions.

We speculate that more facilitative interactions within the microbial community could lead to a more connected network. In such a network, when one species is perturbed, other species are affected since they facilitate each other's growth. Such connections loosen when a microbial community has more inhibitory interactions. Our results suggest that an invader can replace resident species in a community with more inhibitory interactions without disrupting other species. However, in a community with more facilitative interactions, the invader can displace an existing member only if it establishes interactions with other species similar to the replaced species; as a result, displacement is less likely.

## Communities that are more stable are also more resistant to invasion

We hypothesized that loss of richness in resident communities when the community is exposed to invaders is linked to how stable that resident community is when it experiences a perturbation in composition. To examine this hypothesis, we perturbed simulated stable communities such that the Bray-Curtis dissimilarity before and after the perturbation was 0.1. The stable community would undergo two types of perturbations, either the proportion of a randomly selected species was reduced or a new species was added in to emulate exposure to potential invaders. For each simulation, the community was constructed with the parameters shown in Table A in S1 Text. When the composition was perturbed, we tallied the outcomes as either 'robust' when there was no species loss or 'not robust' when some species were lost. For

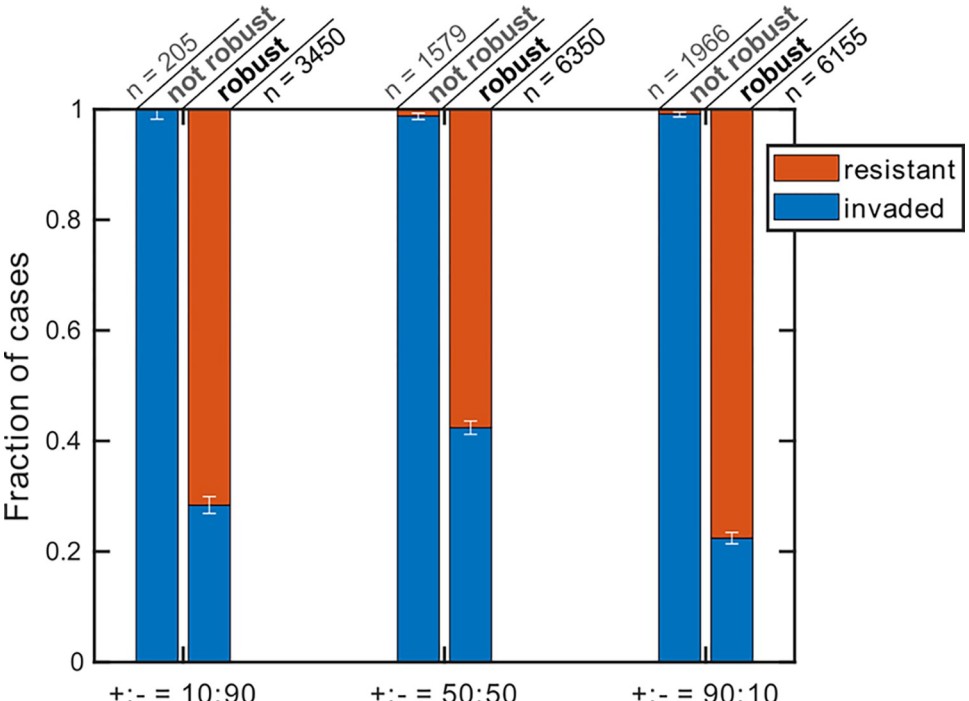

**Fig 2. Robustness against perturbation and resistance to invaders are correlated.** Perturbations of similar magnitude (Bray-Curtis dissimilarity of 0.1) were created either by reducing the proportions of one of the resident members or by introducing an invader. We then assessed among cases that were robust versus not robust when subjected to composition perturbation what fraction resisted or allowed invasion. We tested three conditions: when the microbial interactions between resident members are more inhibitory (+:- = 10:90), equal mix of inhibitory and facilitative (+:- = 50:50), and more facilitative (+:- = 90:10). For each plot, 10,000 instances are examined, and only instances that had two or more stably coexisting species in the resident community were included in the analysis (*n* shows the number of instances in each category).

invasion cases, we tallied how often the community resisted invasion ('resistant') or allowed invasion ('invaded').

We observed that resistance was more pervasive in communities that were robust against perturbations in their composition. In contrast, unstable communities rarely showed resistance to invaders (<2% in each of the +:- = 10:90, 50:50, or 90:10 cases in Fig 2).

## The number of mediators is not the source of the resistance-richness trend

We hypothesized the possibility that with more species a greater number of mediators will be present in the community, facilitating the growth of invaders which could benefit from available resources. We constructed microbial communities starting from 30 mediators in the initial pool and simulated the enrichment until we reached stable communities. The number of mediators in the resulting communities was lower than 30, and only communities with up to 7 mediators had enough instances (>30) to lead to reliable statistics. We then introduced an invader into these communities. Notably, the invasion outcomes showed little correlation with the number of mediators (Fig 3).

## Unexploited mediators affect the relationship between resistance and species richness

Many previous studies attribute the positive correlation between colonization resistance and diversity to the competition of nutrients between species [14,29,31,32]. The justification is that

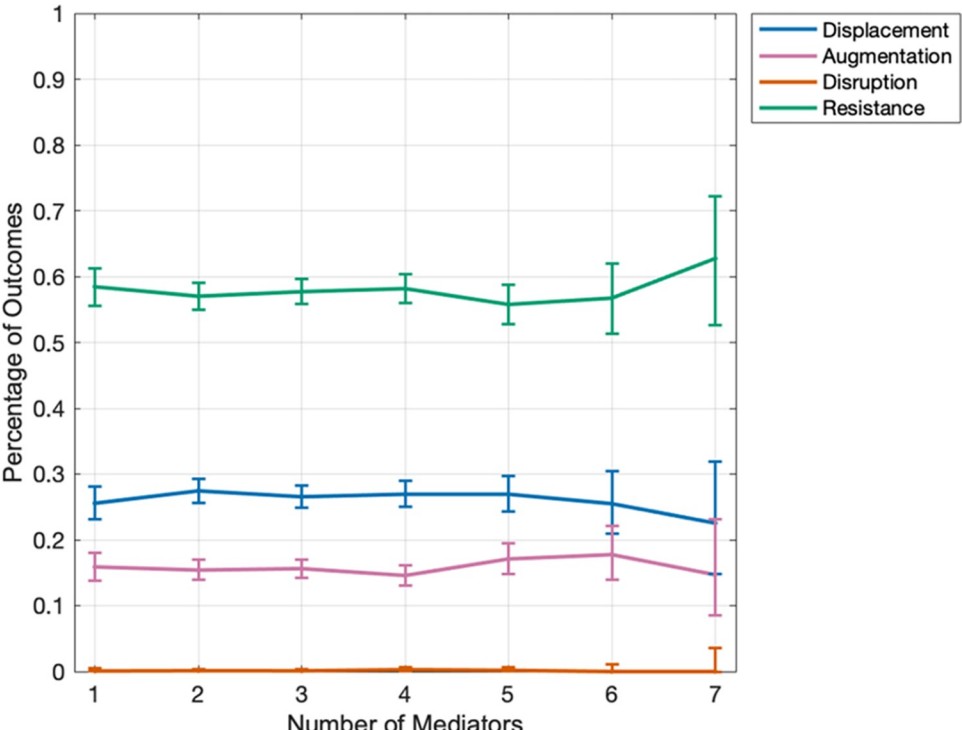

**Fig 3. Invasion outcomes are insensitive to the number of mediators present in the resident community.** Here 10,000 microbial communities have been simulated, each initially containing 20 species and 30 mediators. The invasion outcomes (displacement, augmentation, disruption, and resistance) is plotted with respect to the number of mediators that the community contained when it reached stability. The fraction of interactions between the invader and the mediators that are facilitative ($f_{pI}$) is 0.5 and the initial pool of species for the resident community has an equal mix of inhibitory and facilitative interactions (+:- = 50:50). Only datapoints (in terms of the number of mediators) are shown that had a sample size greater than 30. The error bars show 95% confidence interval.

when a microbial community has more species, they can more efficiently utilize available resources so that little resources are remained for the invader to exploit. To test whether unexploited mediators influence the inverse relationship between invasion outcomes and species richness, we compared the conditions that either the invader utilized the same mediators as the resident community (4a) or an additional mediator was included that was not consumed by the resident community (Fig 4B).

When mediators have been completely exploited (Fig 4A), the relationship between resistance and richness was mainly driven by an increase in displacement instances at higher richness and a decrease in augmentation instances. When the mediators in the environment were not completely exploited by the resident community (Fig 4B), the overall relationship between resistance and richness was the same, with one major difference: more instances of disruption were present at higher richness. The higher abundance of disruption instances leads to a more rapid decrease in instances of resistance in resident communities with higher richness when there are additional mediators to exploit. These trends held when the interactions were more facilitative (+:- = 90:10), more inhibitory (+:- = 10:90), or equally likely to be facilitative or inhibitory (+:- = 50:50).

### The interaction between invader and community influences the resistance-richness trend

Does the facilitation or inhibition effect of the microbial community on the invader influence the inverse relationship between colonization resistance and species richness? We varied the

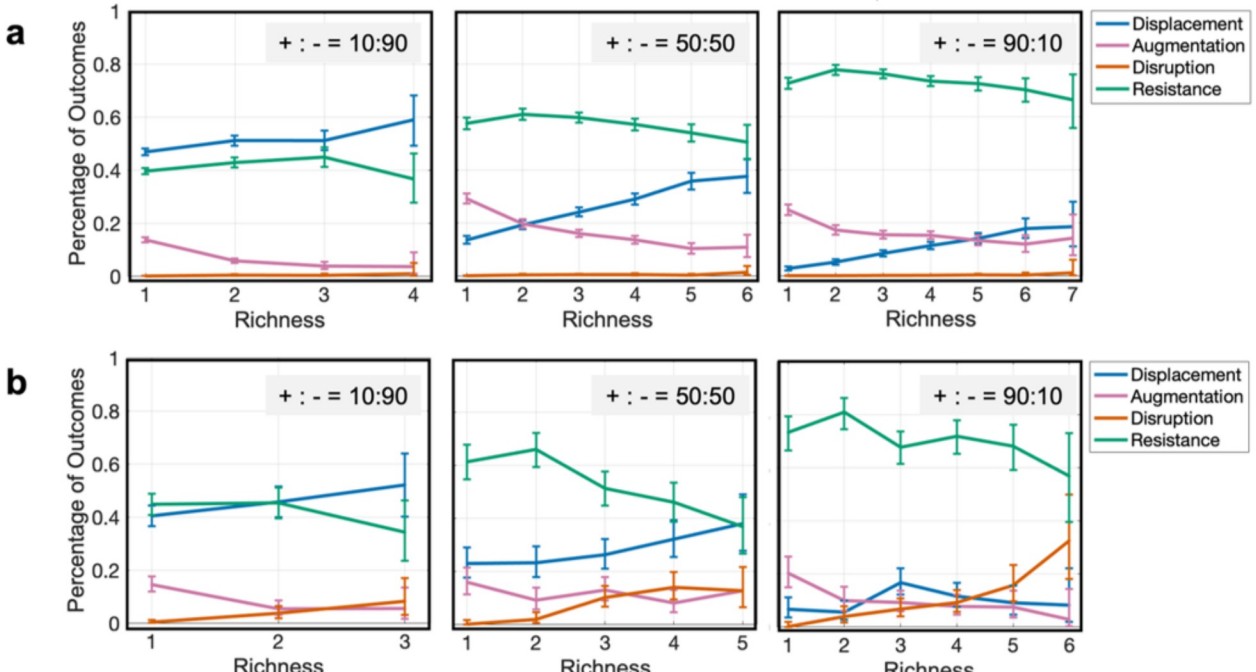

**Fig 4. Unexploited mediators negatively affect resistance.** The relationship between richness and invasion outcomes as a) the invader interact with the same mediators as the resident members or b) the invader produces an additional mediator not present in the resident community. For both cases, three conditions are tested when the microbial interactions between resident members are more inhibitory (+:- = 10:90), equal mix of inhibitory and facilitative (+:- = 50:50), and more facilitative (+:- = 90:10). For each plot 10,000 instances are examined. Data is shown for richness values that have at least 30 instances. The error bars show 95% confidence interval.

fraction of positive interactions between the resident mediators and the invader (parameter $f_{pI}$) to see its effect (Fig 5). In our results, when the interactions between the community and the invader were more inhibitory (i.e., smaller $f_{pI}$), the invasion outcome shifted toward resistance. When the interactions between the community and the invader were more facilitative (i.e., larger $f_{pI}$), displacement replaced resistance. Intuitively, our results show that a community that helps the invader is less capable of resisting the invader, and this effect is stronger with more resident species. Notably, when mediators in the resident community are more

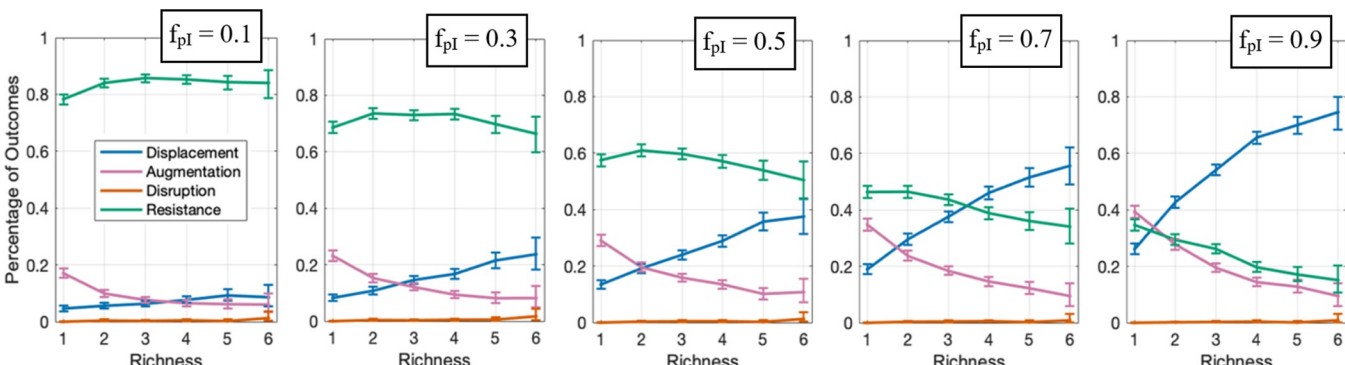

**Fig 5. If the resident community facilitates the invader, the chance of resistance decreases; this effect is stronger when the resident communities has higher richness.** The relationship between richness and invasion outcomes with different $f_{pI}$ (i.e., the fraction of mediator influences on the invader that are facilitative) are shown. The microbial interactions between resident members are equal mix of inhibitory and facilitative interactions (+:- = 50:50). Numbers of instances examined $N_s$ for each plot is 10,000. Data is shown for richness values that have at least 30 instances. The error bars show 95% confidence interval.

facilitative for the invader, instances of displacement increase, which in turn drives the resistance trend such that higher-richness communities are less resistance against invaders.

## Discussion

To investigate how the diversity of microbial communities influences their resistance against invasion, we simulated the introduction of an invader into stable resident microbial communities using the mediator-explicit model of microbial interactions. In our investigation of invasion outcomes, we considered both the maintenance of the invader species and the change in the compositions of the resident species [15]. This is different from most previous studies which usually only track the success of invasion. Such an invader-centric analysis causes two problems [23]. First, the factors that maintain the invader in the system are not necessarily the factors that make invasion possible [23,51]. Second, focusing on the invader ignores whether the resident community stays intact or not [23,43]. Categorizing the four outcomes of displacement, disruption, augmentation, and resistance helps us have a better view of how the invader interacts with the resident community. We investigated the interactions between the resident members, the interaction between the resident members and the invader, the number of initial mediators, and the presence of additional mediators after the invasion to test how these factors affect the richness-resistance relation.

The general trend we observe is that typically the colonization resistance of microbial communities increases with richness in low-richness communities and decreases with richness in high-richness communities (Fig 1), regardless of the details of the network of interactions (Fig 3). Loss of resistance in high-richness communities is partially because these communities are more prone to loss of richness when facing perturbations, whether that perturbation is in the form of a fluctuation in composition or the introduction of an invader species (Fig 2). This trend has also been observed by analyzing human gut microbiome, with DBD more dominant in low-diversity communities and EC more dominant in high-diversity communities [37]. This description can reconcile some seemingly contradictory trends in past reports. Some previous studies suggest a positive relationship between richness and resistance, driven by the niche opportunity theory: a community with higher richness has greater resource partitioning and inhibits invaders more effectively [14,29,31,32]. Other previous studies suggest a negative relationship between richness and resistance, driven by the idea that a more diverse community can also create more niches that can allow an invader to establish. Our analysis offers a simple interpretation that this regime shift can be the consequence of two underlying trends: with increased richness the chance of augmentation decreases and the chance of displacement increases. Note that this interpretation does not exclude other underlying mechanisms, such as niche availability, responsible for the overall trend. Nevertheless, we think making this distinction is useful, because it allows us to target mechanisms that change augmentation or displacement in order to control resistance.

We observe the inverse relationship between the resistance and the richness is more predominant when the interactions between the resident species and the invaders are more facilitative and when the invader introduces a new metabolite to the community. Both of these trends can be explained by underlying patterns of augmentation, displacement, and disruption. Briefly, when the invader benefits from facilitation by the resident species, augmentation and displacement become more prevalent, as the invader is assisted by the community itself to establish (Fig 5). Among these cases, in communities with higher richness displacement can become the dominant outcome. When the invader introduced a new metabolite to the community, an increase in the chance of disruption of high-richness resident community leads to a more pronounced decrease of resistance at higher richness (Fig 4). In general, our results

suggest that when the network is more complex (i.e., the species richness increases), the community is more likely to collapse when new species are introduced.

There are some limitations in the scope of our investigation. First, we assemble stable communities before challenging them with invaders. Such a scheme might miss other processes of community assembly and invasion. For instance, communities might be assembled through succession, in which the timing of introducing an invader can become critical; such a situation is not accounted for in our analysis in this manuscript. Second, although the mediator-explicit model is fairly general, it does not cover all possible microbial interactions. This in turn limits the generality of our conclusions as it would be the case with other choices of models that represent the community. Additionally, each species in our model, including the invader, has a basal growth rate, meaning that the nutrients for their survival are always provided and species are not a competitor for basic nutrients. Therefore, the invasion outcomes are determined by the interactions between species through their production or uptake of metabolites. Species competition for primary resources is de-emphasized in our model. Third, our choice to study well-mixed communities, limits the spatial context of resident communities that we investigate in this report. The impact of other models of microbial interactions (such as a standard consumer-resource model), other community assembly processes (such as succession), and the spatial structure of the environment remain to be investigated in future work.

## Materials and methods

### Mediator-explicit model

We simulate the *in silico* microbial communities using the mediator-explicit model (Eq 1) [15,45]. In this formulation, $C_i$ and $S_i$ are the concentrations of the chemicals and cell densities of the species, respectively. $\beta_{ij}$ and $\alpha_{ij}$ are the production and uptake rates of the chemicals by species, respectively. $r_{i0}$ is the basal growth rates of the cells and $r_{ij}$ is the interaction strength of chemicals on the species. $K_{ij}$ is the saturation rate for the influence of uptake and growth rate influence. For a standard microbial community (unless specified), $\alpha_{ij}$ and $\beta_{ij}$ have uniform distributions between 0.25 and 0.75 fmole/cell per hour and between 0.05 and 0.15 fmole/cell per hour, respectively. $r_{i0}$ for cells in the community has a uniform distribution between 0.08 and 0.12 hr$^{-1}$ and $r_{i0}$ for invader is 0.15. $r_{ij}$ has the amplitude sampling from a uniform distribution between 0 and 0.2 hr$^{-1}$ and the sign is determined by the binomial distribution with a specified probability of positive signs ($f_p$). $f_p$ equals to 0.1, 0.5, and 0.9 when the interactions within the community are more inhibitory, equally inhibitory or facilitative, and more facilitative, respectively, and $f_p$ of the invader is 0.5. $K_{ij}$ has a uniform distribution between 5,000 and 15,000 fmole·ml$^{-1}$. The probability of the presence of links within the community, including the influence of chemicals on species and species' production and uptake of chemicals, is 0.3, and the probability of the presence of links between the invader and the resident species is 0.3.

### *In silico* invasion assay

For each instance of simulation, the initial invasion assay contains 20 species and 10 chemicals, which have pairwise interactions defined by $r_{ij}$, $\beta_{ij}$, and $\alpha_{ij}$. The total number of initial cells is $10^4$ and is evenly distributed to all species. The culture is incubated for a total of 200 generations to reach a stable state. The community is diluted to the same level of cells when the number of cells reaches $10^7$ and the species that has a cell density of less than 10% of its initial density will be dropped out as extinction. Then an invader which also interacts with the chemicals is introduced into the community. The fraction of invader cells is 0.3%. The culture is incubated for another 200 generations following the same procedure. We compare the species richness of resident members before and after the invasion and check whether the invader has

increased composition to categorize the outcomes as either augmentation (species richness retains and invader's composition increases), displacement (species richness decreases and invader's composition increases), disruption (both species richness and invader's composition decrease), and resistance (species richness retains and invader's composition decreases).

## Measures of dissimilarity and diversity

The Bray-Curtis dissimilarity [52] is calculated by

$$BC_{ij} = 1 - \frac{2C_{ij}}{S_i - S_j} \tag{2}$$

where $S_i$ and $S_j$ are the total cell numbers of the community before and after the perturbance, respectively. $C_{ij}$ is the lesser cell number between $S_i$ and $S_j$.

The Shannon Diversity Index [53] is calculated by

$$H = -\sum_{i=1}^{R} p_i \ln p_i \tag{3}$$

where $R$ is the total number of species and $p_i$ is the percentage of each species $i$.

## Simulation platform

All simulations were done in the Matlab R2021a running on the Linux Cluster at Boston College.

## Supporting information

**S1 Text. Fig A.** The chance of four outcomes with respect to Shannon Index under three conditions: the interactions between the resident members a) are more inhibitory ($f_p = 0.2$) b) are equally inhibitory or facilitative ($f_p = 0.5$) c) are more facilitative ($f_p = 0.8$). For each plot 10,000 instances are examined. The Shannon Index of a), b), and c) range between $9.181 \times 10^{-4}$ and 1.049, between $1.710 \times 10^{-4}$ and 1.547, and between $1.685 \times 10^{-4}$ and 1.612, respectively. The Shannon Index range is divided to 30 bins for each condition, and the percentage of each outcome in relation to all instances in each bin range is calculated and plotted. **Fig B.** The overall trends in invasion outcomes obtained using a Lotka-Volterra (LV) model match those of the mediator-explicit model. In these simulations, similar to Fig 1, a pool of $N_{sp} = 20$ species is used as a starting point. The equations used for these simulations were:

$\dot{N}_i = \left[ r_i + \frac{1}{K_i} \sum_j a_{ij} N_j \right] N_i$, where $i$ and $j$ are the species indices, $r_i$ is the species $i$'s growth rate, $K_i$ is the species $i$'s carrying capacity, and $a_{ij}$ is the interaction coefficient. We assume that $a_{ii} = -1$ and that other interaction coefficients $a_{ij}$ are random with a uniform distribution as shown in each panel. The average interaction coefficients is changed from less inhibitory to more inhibitory to examine its impact on invasion outcomes. We simulated the dynamics of this initial pool over 200 generations (20 rounds of growth followed by 1000x dilution) until a stable resident community was reached. The invader was then introduced into the community at a fraction of 0.03% and the outcome was categorized as resistance, disruption, augmentation, or displacement, based on whether the species in the stable community were preserved and whether the invader frequency increased or decreased (as described in Fig 1). Similar to the mediator-explicit model, resident communities with higher richness showed more chance of displacement and less chance of augmentation. This led to an overall nonmonotonic resistance-richness relationship which was more pronounced when the interactions within the community were more inhibitory. For each plot, 50,000 instances of invasion are examined.

Interactions between resident members and the invaders have the same distribution as the interactions among resident members. The error bars show 95% confidence level estimated assuming a binomial distribution for each outcome. Results are shown only at richness value with at least 30 instances. **Fig C.** The overall trends in invasion outcomes obtained using an LV model match those of the mediator-explicit model. All parameters, equations, and assumptions are similar to Fig B, except that the distribution of off-diagonal interaction coefficients $a_{ij}$ has a uniform distribution with a different spread in each panel. The spread of interaction coefficients is changed from less a narrower range to a wider range to examine its impact on invasion outcomes. Similar to the mediator-explicit model, resident communities with higher richness showed more chance of displacement and less chance of augmentation. This led to an overall nonmonotonic resistance-richness relationship which was more pronounced when the interactions within the community were more inhibitory. For each plot, 50,000 instances of invasion are examined. Interactions between resident members and the invaders have the same distribution as the interactions among resident members. The error bars show 95% confidence level estimated assuming a binomial distribution for each outcome. Results are shown only at richness value with at least 30 instances. **Table A**. Parameters used for standard simulations.
(PDF)

## Acknowledgments

We acknowledge computational resources and support offered by Boston College Research Services and the Linux Cluster staff.

## Author Contributions

**Conceptualization:** Yu Zhu, Babak Momeni.

**Data curation:** Yu Zhu.

**Formal analysis:** Yu Zhu, Babak Momeni.

**Funding acquisition:** Babak Momeni.

**Investigation:** Yu Zhu, Babak Momeni.

**Methodology:** Yu Zhu, Babak Momeni.

**Software:** Yu Zhu, Babak Momeni.

**Supervision:** Babak Momeni.

**Validation:** Yu Zhu, Babak Momeni.

**Visualization:** Yu Zhu, Babak Momeni.

**Writing – original draft:** Yu Zhu.

**Writing – review & editing:** Yu Zhu, Babak Momeni.

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
