## [Decision Letter · Decision Letter 0]

8 Mar 2024

Dear Babak,

Thank you very much for submitting your manuscript "Revisiting the invasion paradox: resistance-richness relationship is driven by augmentation and displacement trends" for consideration at PLOS Computational Biology.

As with all papers reviewed by the journal, your manuscript was reviewed by members of the editorial board and by several independent reviewers. In light of the reviews (below this email), we would like to invite the resubmission of a significantly-revised version that takes into account the reviewers' comments.

We cannot make any decision about publication until we have seen the revised manuscript and your response to the reviewers' comments. Your revised manuscript is also likely to be sent to reviewers for further evaluation.

Sincerely,

Ned S. Wingreen

Guest Editor

PLOS Computational Biology

Denise Kühnert

Section Editor

PLOS Computational Biology

Reviewer's Responses to Questions

**Comments to the Authors:**

Reviewer #1: • Abstract and introduction are well written, easy to follow and motivate the study really well.

• The phrase "mediator-explicit" model isn't immediately obvious even to me and I am familiar with much of the invasion literature. Suggest defining this term at the first use -- as a model that is mechanism aware?

• Line 87-88 -- In service of making the paper more forceful I would strengthen the statements here. The motivation is a bit bigger than just to resolve "some of the ambiguity" about "what invation success means." I think the authors have already clearly defined what invasion success means, and here we are trying to disambiguate the seemingly contradictory role of richness/diversity. Encourage authors to sharpen this statement if they wish.

• Line 92 "structural diversity" - demographic analysis. I'm not sure what this means. Probably my own ignorance, but could you clarify what this is? All the other possible quantifications of diversity make sense to me.

• Lines 108-111. I would just include the explicit model here (Eqn. 1A/B from the Methods). This would be clearer -- and also help address my "mediator-explicit" question above.

○ Again at line 185 -- what does "mediators" mean? I think it means nutrients?

○ This ambiguity makes Fig. 4 very hard to understand.

• Also -- can you clarify in the results section how the simulations were done? Was the community first run to a fixed point (stable coexistence under chemostat or serial dilution) and then the invasion was done? I think simple explanations of this would help the reader.

○ This "+:-" notation is confusing.

○ Line 110 -- I don't really know what positive or negative mediator influences mean. Can you explain these in intuitive terms in the context of the model that you used? Competition for shared resources vs. cross feeding of resources seems to be the distinction.

• Minor point -- in the Methods authors refer to C_i as "chemicals" but I think more reasonable to call these either nutrients or resources.

• Fig 1 top -- it would help to make the invader a color that stands out (brighter? Red? Green?) so I can quickly see the instances where invasion succeeds.

• Why does the range of the x-axis varying across the three panels in the bottom of Fig. 1? Is this because for the different interaction profiles it is easier/harder to sustain rich communities?

○ Along these lines -- the decline in resistance fraction for richer communities (green lines Fig.1 right two panels bottom) is modest -- it seems worth considering taking these simulations out to higher richness if this is computationally feasible.

• Paragraph at line 153 is a really nice summary of the main result of this study. Worth emphasizing.

• Line 168 -- there is not Table 1 that I can find.

• The setup to fig. 1 is confusing. The authors measure stability in TWO ways -- reducing the proportion of randomly species OR invading. The fact that the authors including invasion in a measure of stability and then measured invasion resistance for "stable" and "unstable" communities seems tautological. Haven't they defined stability as invasion resistance and then claimed that stable communities are invasion resistant? If the stability assessment of communities only included perturbations to abundances of members of the community there would be no problem here!

• In Figs. 1, 3,4. -- If you chose line colors that were similar for cases where the invader succeeded vs failed this would really help in reading the plots. Alternatively, you could include panels that showed the classic success rate vs richness (irrespective of mechanism) below each panel. This might be nice. Up to the authors.

○ E.g. in line 262 authors are claiming a trend that requires the reader to "add up" two lines in the plots. This is a bit confusing and weakens the results.

• The result in Fig. 5 is nice -- is interesting that increasing the positive interactions with the invaders increases the displacement fraction -- so faciliatory interactions with the invader increase the rate at which members of the community are displaced. Essentially members of the community are helping the invader displace other members of the community. Just a comment.

• Line 280 -- when the invader introduces a new metabolite -- I don't understand under what conditions a new metabolite is introduced by the invader. I think this has to do with my confusion about what a mediator is.

• Line 303 -- the model used here looks like a variant of a CRM to me, so this statement is confusing.

• An aside -- it appears this work was done largely by an undergraduate. Impressive for someone early in their training. Well done!

Reviewer #2: Zhu and Momeni, using a consumer-resource model, explore how diversity and the type of interactions affect the outcome of invasions. Invasion outcomes are not limited to the successful or unsuccessful establishment of the new species, but include also what happens to the resident community, for example whether native species go extinct or not when the invader is successful. The main claim is that, as diversity of the native community increases, augmentation (that is when native species survive the invasion) becomes less likely, while displacement (that is when some native species do not survive the invasion) becomes more likely. Surprisingly, the probability of resistance is not affected by species richness, but it strongly influenced by the fraction of positive interactions.

The paper is well written, but the narrative could be improved. The various simulations could be better connected and figures would benefit from a clear schematic of what is the simulation doing in any particular instance. Some interesting results are presented, but the model the authors used was developed in a previous paper and some of the trends that are discovered are weak.

I have three suggestions:

1. Explain clearly what has been done in the simulations described in “Communities that are more stable are also more resistant to invasion”. From the current version of the text, it is unclear whether after the two possible perturbations (reduction of the biomass of one species and invasion) there is a second invasion.

2. Explain how the outcomes presented in this paper would change when “mediators” are not explicit, like in the case of the Lotka Volterra model

3. I find the word “mediator” instead of resources quite misleading. Why are the authors using this term instead of “resource”?

**Have the authors made all data and (if applicable) computational code underlying the findings in their manuscript fully available?**

Reviewer #1: Yes

Reviewer #2: Yes

PLOS authors have the option to publish the peer review history of their article (what does this mean?). If published, this will include your full peer review and any attached files.

Reviewer #1: No

Reviewer #2: No
---

## [Editor Report · Decision Letter 1]

24 May 2024

Dear Bobek,

We are pleased to inform you that your manuscript 'Revisiting the invasion paradox: resistance-richness relationship is driven by augmentation and displacement trends' has been provisionally accepted for publication in PLOS Computational Biology.

Best regards,

Ned S. Wingreen

Guest Editor

PLOS Computational Biology

Denise Kühnert

Section Editor

PLOS Computational Biology

---

## [Editor Report · Acceptance letter]

7 Jun 2024

PCOMPBIOL-D-24-00129R1 

Revisiting the invasion paradox: resistance-richness relationship is driven by augmentation and displacement trends

Dear Dr Momeni,

I am pleased to inform you that your manuscript has been formally accepted for publication in PLOS Computational Biology. Your manuscript is now with our production department and you will be notified of the publication date in due course.

With kind regards,

Anita Estes
